# Exploring the FGF/FGFR System in Ocular Tumors: New Insights and Perspectives

**DOI:** 10.3390/ijms23073835

**Published:** 2022-03-30

**Authors:** Alessandra Loda, Marta Turati, Francesco Semeraro, Sara Rezzola, Roberto Ronca

**Affiliations:** 1Department of Molecular and Translational Medicine, School of Medicine, University of Brescia, 25123 Brescia, Italy; a.loda025@unibs.it (A.L.); m.turati004@unibs.it (M.T.); sara.rezzola@unibs.it (S.R.); 2Eye Clinic, Department of Medical and Surgical Specialties, Radiological Sciences and Public Health, University of Brescia, 25123 Brescia, Italy; francesco.semeraro@unibs.it

**Keywords:** ocular tumors, FGF, FGFR, retinoblastoma, uveal melanoma

## Abstract

Ocular tumors are a family of rare neoplasms that develop in the eye. Depending on the type of cancer, they mainly originate from cells localized within the retina, the uvea, or the vitreous. Even though current treatments (e.g., radiotherapy, transpupillary thermotherapy, cryotherapy, chemotherapy, local resection, or enucleation) achieve the control of the local tumor in the majority of treated cases, a significant percentage of patients develop metastatic disease. In recent years, new targeting therapies and immuno-therapeutic approaches have been evaluated. Nevertheless, the search for novel targets and players is eagerly required to prevent and control tumor growth and metastasis dissemination. The fibroblast growth factor (FGF)/FGF receptor (FGFR) system consists of a family of proteins involved in a variety of physiological and pathological processes, including cancer. Indeed, tumor and stroma activation of the FGF/FGFR system plays a relevant role in tumor growth, invasion, and resistance, as well as in angiogenesis and dissemination. To date, scattered pieces of literature report that FGFs and FGFRs are expressed by a significant subset of primary eye cancers, where they play relevant and pleiotropic roles. In this review, we provide an up-to-date description of the relevant roles played by the FGF/FGFR system in ocular tumors and speculate on its possible prognostic and therapeutic exploitation.

## 1. Introduction

The eye is a highly specialized sensory organ that allows the collection of external images through photoreception, a process by which light energy is detected by specialized neurons in the retina, i.e., the rods and cones. In turn, retinal neurons activate action potentials, which are subsequently transmitted through the optic nerve to the brain, where the information is processed as vision [1].

Structurally, the eye is a slightly asymmetrical globe located in the orbit, a compartment that is closed medially, laterally, and posteriorly (Figure 1). The eyeball is formed by three concentric layers of tissue. The outer protective layer is constituted by the fibrous coat, which includes the transparent cornea and the opaque sclera; it helps to maintain intraocular pressure and provides an attachment site for intraocular muscles. The anterior portion of the eye and the inner surface of the eyelids are covered by the conjunctiva, a protective mucous membrane [1,2]. The middle layer, i.e., the uvea, represents the vascular coat, which exerts nutritive functions to support ocular structures. It comprises the iris, the ciliary body, and the choroid. Finally, the retina constitutes the neural coat, an inner sensory layer which hosts several classes of neuronal cells involved in the visual process [2,3,4].

The globe is divided into two cavities, the anterior and the posterior segments. The anterior segment encompasses the space around the iris and is filled with the aqueous humor, a clear fluid actively secreted by the ciliary processes. The posterior segment is located behind the lens, and it contains the vitreous humor, which is mostly composed of collagen and hyaluronic acid; vitreous humor has a very slow turnover and it helps in maintaining the shape of the eye [1].

## 2. Ocular Cancers

Among the numerous pathologies that may affect the eye and impair vision, ocular cancers are relatively rare, affecting approximately 1/100,000 in the U.S; their occurrence is variable, according to patients’ ethnicity and age [5,6]. Depending on the type of tumor, ophthalmic malignancies might involve distinct ocular structures (Figure 1) [6]. Moreover, the eye may represent the site of metastasis of other primary tumors such as breast and lung cancers, cutaneous melanoma, tumors of the gastrointestinal tract, and kidney cancer [5,7]. In this review, we focus on the most common intraocular cancers, i.e., retinoblastoma, ocular melanomas, and ocular lymphoma, which together represent the majority of ophthalmic neoplasms.

### 2.1. Retinoblastoma

Retinoblastoma is an ophthalmic tumor that predominantly affects children before 4–5 years of age. It represents the most common intraocular malignancy of childhood and, with approximately 9000 new cases diagnosed each year, it accounts for approximately 2% of all childhood cancers worldwide [8]. Retinoblastomas may occur unilaterally or bilaterally. Unilateral tumors develop following the inactivation in a susceptible retinal cell of both wild-type alleles of the *RB1* gene, which codifies for a regulatory transcription factor. On the other hand, all bilateral patients present a germline mutation of *RB1*; therefore, a second hit is sufficient for the development of the benign precursor retinoma, whereas further mutations are necessary for the progression to retinoblastoma [9]. During the initial stages of the disease, retinoblastoma manifests as a circumscribed intraretinal mass. However, tumors can grow in an exophytic pattern towards the subretinal space, causing diffuse retinal detachment. Alternatively, retinoblastoma can extend in an endophytic pattern within the retina and into the vitreous cavity, leading to vitreous seeding and, in severe cases, infiltrating the anterior segment of the eye [5,10]. If untreated, retinoblastoma is lethal within two years, due to intracranial tumor growth and disease dissemination [11]. Therefore, early diagnosis is essential for its successful clinical management [8].

Currently, chemotherapy combined with focal laser therapy is the preferred method of treatment, while external beam radiation is no longer recommended due to increased risk of secondary malignancies [10]. Enucleation, i.e., the surgical removal of the eye, remains indicated for advanced tumors or in cases of recurrent disease [5]. To date, retinoblastoma has a very high cure rate, with 98% of patients surviving after treatment [10,11]. Nevertheless, metastatic disease occurs in 5% of all retinoblastoma cases and may affect the central nervous system, the bones, and the bone marrow. Despite the successful treatment of the primary tumor, the prognosis for metastatic retinoblastoma remains poor, and few therapeutic options are available [8,10].

### 2.2. Ocular Melanomas

Ocular melanomas are the second most common type of eye tumor, and they represent 10% of all melanomas. They arise from the melanocytes located in different regions of the eye, mainly within the uvea and the conjunctiva, giving rise to uveal or conjunctival melanomas, respectively [12].

Uveal melanoma is the most frequent primary intraocular neoplasm in the adult population, and it accounts for 85% to 95% of all intraocular melanomas [13]. The incidence of uveal melanoma in Europe ranges from 2 to 8 cases per million, and its occurrence increases with age [14]. Tumors may originate from any region of the uveal tract, which is composed of the iris, the ciliary body, and the choroid. Clinical presentation varies according to tumor size and location, with blurred and distorted vision being common symptoms of iris or ciliary body involvement, while choroidal melanomas are associated with vision loss due to retinal detachment [15]. Primary tumors are successfully treated by brachytherapy or phototherapy, whereas enucleation is recommended only for severe cases with extensive intraocular growth [16]. Nevertheless, uveal melanoma is a highly metastatic disease, with a tendency to spread via hematogenous dissemination; the liver represents the most frequent site of metastasis, followed by lungs, bones, skin, and brain [15,17]. Approximately 50% of patients affected by uveal melanoma develop metastasis within 5 years, with median survival ranging from 4 to 15 months due to the lack of effective pharmacological therapies [18]. To date, no standard of care has been approved for treatment of metastatic disease and conventional chemotherapies remain unable to improve the overall survival [14,19,20]. Thus, novel therapeutic approaches are eagerly required. In this context, in vitro and in vivo experimental models of uveal melanoma may represent a useful tool for the screening of new drug candidates [21]. It is worth mentioning that uveal melanoma lacks the most typical mutations associated with cutaneous melanoma (i.e., *BRAF* and *NRAS*). Instead, activating mutations in *GNAQ* or *GNA11* genes occur in 80–90% of uveal melanoma cases in a mutually exclusive pattern. Both *GNAQ* and *GNA11* genes codify for α-subunits of G-coupled proteins and have been recognized as uveal melanoma driver mutations [22]. Additionally, mutations of *BAP1* are frequently observed in most metastasizing uveal melanomas. Loss of *BAP1* compromises the maintenance of a differentiated melanocytic phenotype, promoting epithelial-to-mesenchymal transition and metastatic dissemination. In alternative to *BAP1*, metastatic uveal melanomas often present mutations of *SF3B1*, which are associated with a longer disease-free survival [13].

Conjunctival melanoma comprises approximately 5% of all ocular melanomas and it arises from melanocytes located in the basal layer of the conjunctival epithelium, which lines the eyelids and the sclera [12,23]. In rare cases, tumors might grow and extend toward the orbit or into the globe. Moreover, conjunctival melanoma tends to spread via both lymphatic and blood vessels, affecting first the regional lymph nodes in 45% to 60% of patients [12]. Subsequent systemic dissemination may occur in 20% to 30% of patients within 10 years, with metastasis spreading to lungs, brain, bones, liver, skin, and the gastrointestinal tract [12,23]. Currently, the standard treatment for conjunctival melanoma is the surgical resection of the tumor mass, followed by cryotherapy to the tumor margins after excision. However, effective eradication of conjunctival melanoma is hindered by a high rate of local recurrence. Therefore, adjuvant chemotherapy is often employed through the administration of topical agents [24]. Enucleation or orbital exenteration, which consist in the surgical removal of the globe, muscles, nerves, and fatty tissue adjacent to the eye, may be necessary for patients with advanced tumors, while no standard of care has been defined for metastatic disease [12,24,25].

### 2.3. Ocular Lymphomas

Intraocular lymphomas are a rare type of malignant lymphocytic neoplasm and they include lymphomas derived from the vitreoretinal tissues as well as lymphomas of the uveal tract. Vitreoretinal lymphomas are mainly primary diseases, arising within the central nervous system, while uveal lymphomas generally occur as metastasis of systemic non-Hodgkin lymphomas [26,27]. The exact epidemiology of primary intraocular lymphomas is unclear, as most datasets classify the disease as a subset of primary central nervous system lymphomas [27]. Tumor onset is often subtle, with non-specific symptoms that mimic uveitis and lead to a delayed diagnosis [26]. Moreover, 16% to 34% of patients also manifest central nervous system involvement at presentation. Indeed, the disease progresses to intracranial lymphoma in 42% to 92% of patients, with widespread dissemination occurring in the advanced stages of the disease [27]. Optimal treatment for intraocular lymphoma is not well defined. The primary disease is mainly treated by intravitreal chemotherapy or low-dose localized radiotherapy, whereas high-dose chemotherapy combined with local therapy is recommended for patients with central nervous system involvement [26,28]. Mortality rates are inconsistent due to the rarity of the disease, spanning from 9% to 81% in follow-up periods of 12–35 months [27].

### 2.4. Eye Metastasis

The eye may represent the site of metastasis for several tumors, in particular breast (47%) and lung (21%) cancers, but also of cutaneous melanoma, tumors of the gastrointestinal tract, and kidney cancer [5,7]. Metastasis might arise in any part of the eyeball or the orbit, but 88% of cases affect the posterior uvea due to its extensive vascularization [7,29]. Therapeutic strategies include systemic therapies, local treatment, or a combination of both [29]. Radiotherapy, either with external beam radiation or brachytherapy, is the most common treatment for metastatic disease [30]. However, the average survival expectation following diagnosis of ocular metastasis is approximately 7 months and is essentially linked to the lethality and stage of the primary tumor [7].

## 3. The Fibroblast Growth Factor (FGF)/FGF Receptor (FGFR) System

In mice and humans, the Fibroblast Growth Factor (FGF) family is composed of 22 polypeptides that act as secreted signaling proteins (FGF1-10, FGF16-23) or as receptor-independent intracellular factors (FGF11-14), with the latter being mainly involved in neuronal development and in regulating the electrical excitability of neurons [31,32]. Secreted FGFs are grouped into 6 subfamilies according to phylogenetic analysis and sequence homology. The subfamilies FGF1/2/5, FGF3/4/6, FGF7/10/22, FGF8/17/18, and FGF9/16/20 are known as canonical FGFs and act as local paracrine signaling molecules. The FGF19/21/23 subfamily comprises hormone-like FGFs acting as endocrine factors that control metabolic homeostasis [31,33,34]. Both canonical and hormone-like FGFs mediate their biological functions by activating cell surface tyrosine kinase (TK) receptors (FGFRs), which are encoded by four distinct genes (*FGFR1-4*) in mammals [31,33]. Structurally, FGFRs present an extracellular domain, a transmembrane domain, and a cytoplasmic TK tail, which is responsible for FGF-related signaling. The extracellular domain consists of three immunoglobulin (Ig)-like domains (I–III), with the Ig-like domain II and III being involved in ligand binding and in defining ligand specificity [33,35]. The functional interaction between canonical FGFs and their receptors requires the formation of two FGF-FGFR-heparan sulfate proteoglycan (HSPG) ternary complexes and their subsequent dimerization (Figure 2) [33,36].

Besides their role as coreceptors in FGF/FGFR interaction, HSPGs protect canonical FGFs from extracellular protease-mediated degradation; moreover, they sequester FGF molecules, thus limiting their diffusion through the extracellular matrix and providing a reservoir of the ligands [35,37]. The formation of the FGF-FGFR-HSPG ternary complex triggers conformational changes, leading to trans-phosphorylation of the tyrosine residue within the intracellular TK domain and providing docking sites for intracellular receptor substrates, such as specific adaptor protein FGFR substrate 2 (FRS2) and phospholipase Cγ (PLCγ). Phosphorylation of FRS2 activates the RAS-MAPK pathway, resulting in proliferation, differentiation, or cell cycle arrest, depending on the different cellular context. Moreover, FRS2 phosphorylation may also activate the anti-apoptotic PI3K-AKT pathway. On the other hand, PLCγ leads to protein kinase C (PKC) activation and intracellular Ca^2+^ release, promoting cell migration [34,38] (Figure 2).

By mediating such a wide range of cellular activities, the FGF/FGFR system assumes pivotal regulatory roles. Indeed, it is involved from the earliest phases of embryonic development by taking part in mesoderm patterning; moreover, by regulating mesenchymal-epithelial communications, the FGF/FGFR system is essential for organogenesis. Furthermore, FGFs/FGFRs exert homeostatic functions in adults, being involved in tissue repair and remodeling processes [31,34].

Given its ubiquitous and wide-ranging biological functions, the FGF/FGFR system requires tight regulation. Ligand-receptor binding specificity and spatio-temporal expression of FGFs, FGFRs, and HSPGs are necessary to avoid aberrant or unappropriated activation. Furthermore, negative feedback mechanisms occur in response to FGF/FGFR activation, including FGFR internalization and the recruitment of phosphatases and/or negative modulators (e.g., Sprouty proteins) [33,38]. FGFR signaling may also be modulated though the interaction with the non-canonical signaling partners of FGFRs, including extracellular matrix (ECM)-associated proteins, cell adhesion molecules (CAMs), or other transmembrane proteins and serine/threonine kinases [39].

### 3.1. The FGF/FGFR System in Cancer

The FGF/FGFR family has been described to play a relevant role in several pathological conditions, including cancer [33,34,40]. The aberrant activation of the FGF/FGFR system, both in the neoplastic and the stromal compartments, may occur both in a ligand-independent or a ligand-dependent manner, triggering tumor growth, invasion, angiogenesis, metastatic dissemination, and resistance to therapies [41,42,43]. Activating mutations in the extracellular or TK domains of the receptors are involved in the progression of various tumor types, including bladder and cervical cancers [44], multiple myeloma [45], and prostate cancer [46]. Moreover, chromosomal translocations may generate fusion proteins involving the TK domain of FGFR combined with a transcription factor domain, as, for example, ZNF198 in myeloproliferative syndrome [47] or ETV6 in peripheral T-cell lymphoma [48]. In these cases, the constitutive dimerization and activation of the fusion protein strongly promotes cell proliferation and tumor growth [47,48]. As reported for multiple myeloma, chromosomal translocations may also result in *FGFR* overexpression by bringing *FGFR* genes under the control of a highly active promoter [37,38,40,49]. Additionally, *FGFR* overexpression has been reported for breast [50], gastric [51], and squamous cell lung cancers [52] as a consequence of dysregulated gene transcription and amplification.

Ligand-dependent FGFR signaling activation plays an important role in the pathogenesis of cancer as well. Indeed, FGFs can be produced at high concentrations or “out of context” by cancer cells or by the surrounding stroma, thus causing the hyperactivation of the signaling and sustaining tumor growth through autocrine/paracrine mechanisms. Furthermore, altered gene splicing mechanisms may lead to the production of different splice variants of the receptors, able to bind a wider range of FGFs, resulting in an increased FGF/FGFR activation. Aberrant FGF/FGFR signaling may also result from the impairment of negative feedback mechanisms, including mutations that increase receptor stability or loss of negative feedback regulators [37,49].

Besides their pro-tumor activity exerted on cancer cells, tumor-derived FGFs also mediate tumor/stroma crosstalk, thus playing a relevant role in conditioning the surrounding stromal cells and favoring the onset of a pro-tumor microenvironment [53,54]. It is well documented that FGFs, in particular FGF1 and FGF2, promote tumor-associated angiogenesis and induce the formation of new vessels that provide oxygen and nutrients, and that facilitate cancer cell dissemination [49]. Furthermore, tumor-derived FGFs activate cancer-associated fibroblasts (CAFs), and in turn CAF-produced FGFs sustain cancer progression [55]. FGFs are also involved in the recruitment of tumor-associated macrophages, which exert pro-tumor functions by negatively regulating immune responses to cancer cells. Finally, emerging evidence highlights a possible role of the FGF/FGFR system in the acquisition of resistance to drugs, despite their different molecular structure and mechanisms of action [49,56]. Thus, aberrant activation of FGF/FGFR signaling may have several effects on tumor biology, including the promotion of cell proliferation and survival, motility and invasiveness, metastatic dissemination, tumor escape from immune control, and resistance to therapy.

Finally, the regulation exerted by non-canonical FGFR interactors plays a relevant role in cancer. Indeed, integrin-regulated FGFR signaling has been directly implicated in tumorigenesis, particularly in angiogenesis, a critical step for metastatic dissemination. FGF1/Integrin-αVβ3/FGFR1 crosstalk has been shown to promote both angiogenesis and tumorigenesis, and to enhance epithelial to mesenchymal transition (EMT) in breast cancer cell lines [57,58]. FGFR can also interact with different glycoproteins belonging to the family of CAMs, which are strictly implicated in fostering the migratory properties associated with EMT in cancer. Indeed, neural-CAM (NCAM) has been reported to prevent the binding of FGF to its receptor by acting as a nonconventional ligand of FGFR1, able to mediate an FGF-independent activation [59]. NCAM/FGFR1 complexes cycle rapidly and repeatedly at the cell surface and result in sustained signaling and cell migration. Similarly, L1CAM was described to induce signal transduction through FGFR1 in glioma cells, promoting proliferation and motility [60]. FGFR/cadherins interactions have been reported, leading to different biological effects, either tumorigenic or tumor suppressive, depending on the type of cadherin involved [61]. For instance, the binding of N-cadherin with FGFR1 stabilizes the receptor at the plasma membrane, preventing its internalization and degradation, thus promoting motility, invasion, and metastasis [62]. Galectin-1 and -3 have been described to interact with the extracellular regions of FGFR1, mimicking the ligands in an FGF-independent way and acting as regulators of FGFR1 signaling and trafficking [63]. Indeed, FGFR1/galectin-1 complexes trigger the dimerization of the FGFR, the activation of the downstream signaling, and result in anti-apoptotic and proliferative responses [64]. Conversely, galectin-3 crosslinks FGFR1 on the cell surface and prevents its constitutive internalization.

### 3.2. FGF/FGFR Inhibitors

Due to its crucial role in cancer progression, the FGF/FGFR system represents an attractive target for the development of anti-tumor drugs. In this context, FGFR inhibitors may act either at an extracellular level, by preventing ligand-receptor interaction, or at an intracellular level, by hampering signal transduction. Currently, FGFR inhibitors are classified as: (i) TK inhibitors (TKIs), (ii) monoclonal antibodies (mAbs), and (iii) FGF traps [41,49].

First-generation TKIs are small molecules that inhibit the kinase activity of TK receptors (RTKs) by preventing the binding of ATP to the catalytic site in a non-selective manner. These compounds act on several RTKs, including FGFRs, due to the structural similarity of their TK domains [65]. Although simultaneous inhibition of multiple RTKs may represent a compelling therapeutic strategy, the application of non-selective TKIs in clinical practice is limited by the onset of local and systemic complications, together with the poor efficacy observed in FGFR-dependent tumors. Nevertheless, some of these compounds are currently under investigation in preclinical and clinical trials, whereas other non-selective TKIs have already been approved for the treatment of metastatic thyroid cancer (i.e., lenvatinib) and metastatic colorectal cancer (i.e., regorafenib) [41]. To overcome the off-target effects of first generation TKI drugs, selective FGFR TKIs have been developed and are now under evaluation (e.g., BGJ398 for non-muscle-invasive urothelial carcinoma and AZD4547 for non-small cell lung cancer) or already approved (e.g., pemigatinib for cholangiocarcinoma and JNJ-42756493 for urothelial carcinoma) [66,67,68] (www.clinicaltrials.gov, accessed on 17 February 2022).

While most of the compounds described above exert their activity on more than one FGFR, anti-FGFR mAbs have the advantage to target specific receptors or even isoforms. Moreover, they are associated with a reduced toxicity due to the absence of off-target effects. Nevertheless, to date, only two anti-FGFR mAbs have entered clinical trials, i.e., MGFR1877S for the treatment of advanced solid tumors and FPA144 for gastric cancer [65,66] (www.clinicaltrials.gov, accessed on 17 February 2022).

Finally, FGF-trap inhibitors may represent a compelling therapeutic strategy for tumors driven by an aberrant ligand-dependent activation of the FGF/FGFR system. These drugs can bind one or more FGFs and, by acting at the extracellular level, they can also affect the tumor microenvironment, hampering the tumor-stroma crosstalk [41,65]. The FGF-trap family comprises several compounds, including FP-1039, a soluble decoy receptor fusion protein, and NSC12, a small molecule that mimics the minimal FGF2-binding sequence of the long Pentraxin-3 [49,69]. Interestingly, this new class of small molecules has displayed a low toxicity profile when evaluated in experimental animal models [69].

## 4. The FGF/FGFR System in Eye Tumors

Even though the involvement of FGFs/FGFRs has been well documented in most solid and hematological tumors, to date, scattered pieces of literature show that they may also play a relevant role in eye tumors, particularly in uveal melanoma and retinoblastoma.

Clinical and experimental evidence suggests the presence of an FGF/FGFR autocrine activation loop in uveal melanoma. Indeed, data mining performed on the publicly available mRNA profiling dataset of 80 primary human uveal melanoma specimens, present in The Cancer Genome Atlas (TCGA), reports the overexpression of one or more *FGFs* or *FGFRs* in 60% and 21% of cases of uveal melanoma, respectively (Figure 3A). Interestingly, among several *FGFs* and *FGFRs* that were found upregulated, *FGF12* and *FGFR1* were the most represented, reaching 26% and 11% of total cases (Figure 3B). In addition, alterations in *FGFs* and *FGFRs* resulted in a poorer prognosis in terms of reduced overall survival in patients (Figure 3C and [70]). Expression analysis in a set of 9 primary uveal melanomas reported that *FGF1* and *FGF2* were expressed in 77% of samples, with co-expression of *FGF1*/*FGF2* in 55% of cases. Moreover, primary tumors also expressed all *FGFRs*, with *FGFR1* being the most represented overall, while 33% of tumors expressed both *FGF1*/*FGF2* ligands and all four receptors [71].

Clinically, high levels of FGF2 were detected in mixed/epithelioid specimens, associated with a poor prognosis, compared to spindle cell type tumor samples [72,73]. Accordingly, primary tumors expressing *FGF2* were associated with an increased metastasis occurrence [72]. In this context, the elevated expression of *FGF2* in uveal melanoma metastases further reinforces the hypothesis that FGFs play a non-redundant role in uveal melanoma progression and invasion. Indeed, it has been recently reported that FGF2, produced by liver stellate cells, can mediate FGFR activation in metastatic uveal melanoma cells; moreover, it is responsible for the resistance to the bromodomain and histone deacetylase inhibitors [74].

From the perspective of therapeutic applications, the blocking of endogenous FGF2 with monoclonal antibodies or antisense nucleotide reduced cell proliferation, clonogenic potential, and cell survival in uveal melanoma cell lines [71]. Indeed, similar results were obtained by targeting FGFR1 [71]. Accordingly, treatment with the pan FGF-trap NSC12 [69] prevented the activation of FGFRs and their downstream signaling mediators FRS2 and ERK1/2 in uveal melanoma cells [70]. Moreover, NSC12 treatment induced cell apoptosis through the activation of the pro-apoptotic caspase-3 protein as well as PARP cleavage [70]. These events were matched by the degradation of β-catenin, a key mediator of uveal melanoma metastasis [75,76,77], and resulted in a significant inhibition of cell proliferation and migration [70]. Notably, similar effects were obtained with the selective FGFR TK inhibitor BGJ398 [70].

Regarding other ocular neoplasms, scattered evidence obtained on human retinoblastoma cell lines showed the expression of all four *FGFRs*, with cell proliferation in response to stimulation with FGF1 and FGF2 [78,79]. In addition, analysis of aqueous humor from retinoblastoma patients revealed higher concentration of FGF2 compared to the control group, thus supporting the hypothesis that FGF may play a role in retinoblastoma progression [80]. Moreover, experimental evidence shows that treatment with exogenous FGF1 induces the activation and phosphorylation of FGFR1 in the human retinoblastoma Y-29 cell line, while the selective inhibition of FGFR1 resulted in decreased cell proliferation [79].

The activation of the angiogenic switch, which requires an imbalance between pro- and anti-angiogenic factors, is essential for tumor progression [81]. In uveal melanoma and retinoblastoma, an increased vascular density has been associated with larger and more invasive tumors as well as with a poorer prognosis in patients [82,83]. In this frame, high levels of Vascular Endothelial Growth Factor (VEGF) have been reported in the ocular fluids of patients affected by both uveal melanoma or retinoblastoma [80,84,85]. Moreover, a significant reduction of tumor growth was observed following treatment with anti-VEGF bevacizumab, in both in vitro and in vivo experimental models, suggesting that anti-angiogenic strategies may be of significance for the clinical management of ocular tumors [86,87]. Given the role of FGF2 as a potent pro-angiogenic mediator, several studies have investigated its involvement in ocular tumor-associated angiogenesis. As mentioned above, high concentrations of FGF2 have been found in the aqueous humor of patients affected by either retinoblastoma or uveal melanoma [80,85]. Moreover, immunohistochemistry analysis of uveal melanomas showed that, even though FGF2 is mainly located in the cytoplasm of tumor cells, a positive signal is also detectable in the perivascular area [88]. Accordingly, in vitro experiments reported a significant impairment in the proliferation of endothelial cells co-cultured with primary human uveal melanoma cells following the selective inhibition of FGF2 [88], thus pointing to this pathway as a possible target to block neo-angiogenesis in uveal melanoma. Similar results were obtained in a transgenic mouse model of retinoblastoma, where a time-course analysis of FGF2 expression showed a peak of production during the early stages of tumorigenesis, localized in the perivascular area [78]. Accordingly, immunofluorescence analysis of human retinoblastoma tissues showed a positive staining for FGF2 located in both tumor and vascular cells [78]. Finally, Y-29 cells extracts induced proliferation of bovine brain-derived capillary endothelial cells, whereas their pro-angiogenic activity was prevented in the presence of neutralizing anti-FGF2 antibodies [89].

## 5. Concluding Remarks

In the era of personalized medicine and targeted therapies, it is of growing importance to deepen our knowledge on the molecular mechanisms involved in tumor progression; currently, new therapeutic approaches are being constantly investigated and developed. In this context, the FGF/FGFR system represents a paradigm, given its regulatory role in multiple hallmarks of cancer biology, such as proliferation, EMT, angiogenesis, metabolism, and drug resistance. As described in this review, the activity of the FGF/FGFR system has been widely characterized in several tumor types, leading to the introduction of novel therapies, both in clinical trials and in clinical practice [41,90].

Despite their relatively low incidence, eye tumors represent a challenging context for the development of new pharmacological treatments aimed at improving the overall survival of patients as well as their quality of life. In this frame, the FGF/FGFR signaling pathway represents an exploitable therapeutic target, due to its involvement in promoting tumor progression and dissemination, both in uveal melanoma and retinoblastoma. Indeed, experimental data suggest that targeting FGFR deeply affects tumor cells, impairing their capacity to grow, invade, and, eventually, resist first line therapies. Moreover, inhibition of the FGF/FGFR system may also be significant as an anti-angiogenic strategy, taking into consideration the importance of angiogenesis and hematogenous dissemination in ocular tumors, which develop in deeply vascularized area. Despite the lack of direct reports on the pro-angiogenic effect of FGF in ocular tumors, it is reasonable to assume that its mere expression contributes to sustaining neo-vessel formation. Notably, anti-FGF approaches have been widely characterized as anti-angiogenic; furthermore, they can be employed to overcome resistance to conventional anti-VEGF therapies [91,92]. Interestingly, the anti-angiogenic effect exerted anti-FGFs/FGFRs should be considered from the perspective of an integrate approach, aimed at treating ocular tumors by acting on both the stromal and the parenchymal compartments.

In addition to the direct role of the FGF/FGFR system in ocular tumors, further research is required to investigate the activity of the non-conventional FGFR interactors. For instance, the expression of NCAM has been reported in mixed/epithelioid uveal melanoma cell types, which are associated with an increased metastatic potential [93]. Nevertheless, the involvement of NCAM and other FGFR- activators is largely unexplored in the field of ocular neoplasms.

To date, different therapeutic strategies allow us to block FGFRs with a more or less selective approach; however, their clinical application has been reserved only for those tumors where the driving role of FGFRs is well characterized, such as cholangiocarcinoma and urothelial cancers [66,67,68]. Notably, a finer modulation of FGFR activation may be achieved through FGF-trap molecules; therefore, their validation would allow better regulation of the crosstalk exerted by different FGFs in the complex tumor microenvironment [38]. FGF/FGFR inhibitors represent an attractive therapeutic perspective for ocular tumors, and especially for the clinical management of uveal melanoma; nevertheless, their legitimation is hampered by the scarcity of literature reporting their involvement in the different phases of tumor growth. Therefore, more studies are needed to expand the knowledge of the FGF/FGFR system into other, less represented, tumors of the eye and to push the currently available FDA-approved anti-FGF/FGFR drugs towards their application in ophthalmic neoplasms.

## Figures and Tables

**Figure 1 ijms-23-03835-f001:**
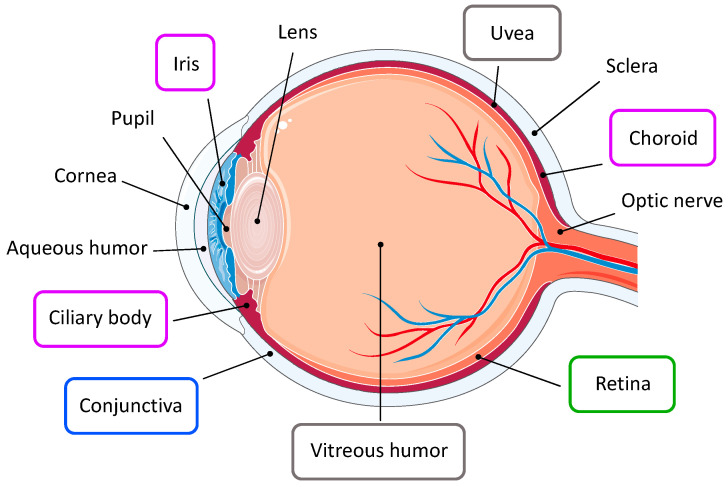
Tumors of the eye. Ophthalmic tumors affect specific ocular structures. Retinoblastoma (green) arises in the retina; conjunctival melanoma (blue) involves the conjunctival epithelium; uveal melanoma (purple) develops from any region of the uveal tract; ocular lymphomas (grey) derive from the vitreoretinal tissue or from the uvea.

**Figure 2 ijms-23-03835-f002:**
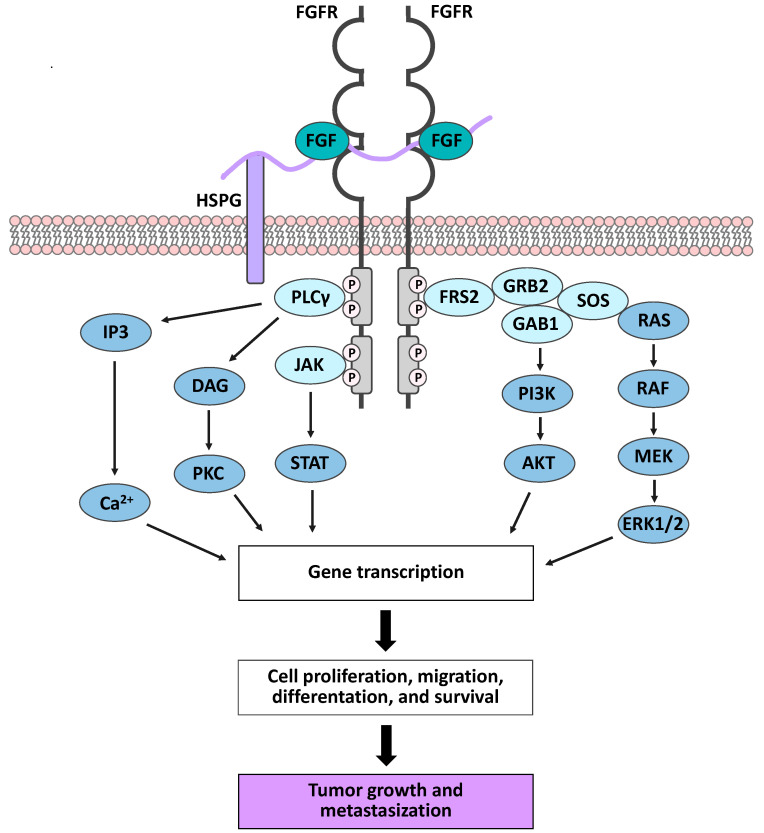
Fibroblast Growth Factor (FGF)/FGF receptor (FGFR) signaling pathways. The formation of two FGF-FGFR- heparan sulfate proteoglycan (HSPG) ternary complexes induces receptor dimerization and trans-phosphorylation of the tyrosine kinase (TK) domains. This event leads to the docking of intracellular receptor substrates and consequent activation of downstream signaling pathways. Deregulation of FGF/FGFR-mediated cell activities promotes tumor onset and progression.

**Figure 3 ijms-23-03835-f003:**
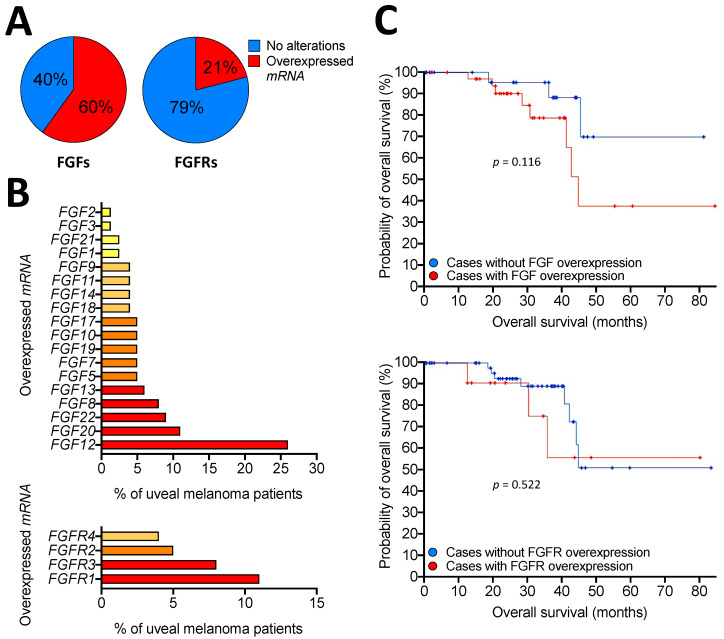
Overexpression of FGFs and FGFRs in human primary uveal melanoma. Analysis of The Cancer Genome Atlas (TCGA) dataset performed on 80 primary human uveal melanoma specimens. (**A**) Pie charts showing the percentage of samples with mRNA overexpression of *FGFs* (left panel) or *FGFRs* (right panel). (**B**) Percentage of uveal melanoma patients with mRNA overexpression of different members of the FGF (upper panel) or FGFR (lower panel) families. (**C**) Probability of overall survival of patients with or without FGF (upper panel) or FGFR (lower panel) alterations. Statistical analysis: Logrank Test.

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
