# Peer review of "Exploring the FGF/FGFR System in Ocular Tumors: New Insights and Perspectives"

_ijms, 2022, doi:10.3390/ijms23073835_

Round 1

Reviewer 1 Report

The manuscript by Loda et al focuses on the role of FGF/FGFR signaling in ocular tumors. The Review manuscript is well written and highlights the role of FGF/FGFR signaling units in highly important, yet a bit inscrutable research field of ocular tumors. The paper is scientifically sound and contains relevant information. I would just propose to expand two parts in the manuscript: it is well known that numerous partner proteins may modulate FGF/FGFR activity in distinct tumors (e.g. NCAM, integrals, galectins) - in my opinion expanding this part and relating it to ocular tumors could be helpful. Similarly, there are number of therapeutical approaches targeting FGF/FGFR developed for distinct tumors (e.g. breast or lung cancer: ADCs, ligand-drug conjugates, ligand traps, kinase inhibitors, PROTACs) - relating these findings in light of their usefulness for ocular tumors would be beneficial for the completeness of this manuscript.

Reviewer 2 Report

The presented publication contains information about the significance of the FGF/FGFR system in ocular tumors. This topic is relevant, since there are few publications on this topic. First of all, it should be noted that earlier this team successfully published an article on this topic in the journal Cancers doi: 10.3390/cancers11091305, which is referred to in this publication.
There are several comments on the material presented:
1. First of all, the review pays little attention to new insights and perspectives proper – most of the text describes fairly well-known things about the types of ocular tumors and the FGF/FGFR-dependent signaling pathway.
2. The prospects for a therapeutic approach based on FGF/FGFR inhibition should be presented in more detail, this section currently includes very little information. For example, are FGFR inhibitors promising?
3. The significance of tumor angiogenesis for the development of ocular tumors and the role of the FGF/FGFR-dependent signaling pathway should be presented in more detail.
4. Concluding remarks and possible prospects for antitumor therapy are written very briefly.

Round 2

Reviewer 2 Report

Thank you for the corrections you have made. The article can be published in its present form